# Effects of Temperature and Salt Stress on the Expression of delta-12 Fatty Acid Desaturase Genes and Fatty Acid Compositions in Safflower

**DOI:** 10.3390/ijms24032765

**Published:** 2023-02-01

**Authors:** Dandan Li, Kaijie Li, Guangchong Zhou, Songtao He

**Affiliations:** Agronomy College, Guizhou University, Huaxi, Guiyang 550025, China

**Keywords:** *Carthamus tinctorius* L., Asteraceae, temperature stress, salt stress, Δ^12^ fatty acid desaturase

## Abstract

The regulation of microsomal (e.g., FAD2) and plastidial (e.g., FAD6) oleate desaturases by cold, heat and salt stress were investigated. Gene expression levels and fatty acid compositions were determined in the roots, stems and leaves of safflower following stress treatments. A safflower plastidial oleate desaturase gene, *CtFAD6*, was cloned, and oleic acid desaturation was confirmed in *Synechococcus* sp. strain PCC7942. The results showed that temperature regulated oleate desaturation at the transcriptional level, and this regulation pattern was tissue-specific. *CtFAD2-1*, *CtFAD2-2* and *CtFAD6* were significantly induced under cold and heat stress in young leaves, and *CtFAD2-2* and *CtFAD6* were slightly induced in young stems. In contrast, *CtFAD2-1*, *CtFAD2-11* and *CtFAD2-10* were sensitive to salt stress in all safflower tissues (roots, stem and leaves). *CtFAD6* was insensitive to salt and was slightly induced in leaves only.

## 1. Introduction

Safflower (*Carthamus tinctorius* L.) belongs to the Asteraceae family and is a multipurpose plant. Safflower is a traditional Chinese medicine (TCM) [1]. Most of the medicinal effects of the extract come from its flowers [2]. Safflower is also an ancient oilseed crop because of its high-quality esculent oil [3]. Safflower has been widely cultivated for its multipurpose values and its cold-, drought- and salt-tolerant characteristics [3]. Safflower seedlings in the rosette stage have good adaptability to temperature change and can tolerate even −5 °C [3]. In Sichuan, China, safflower is usually sown in late autumn, and the optimum growth temperature is 15~18 °C during the seedling stage [3].

Linoleic acids (18:2) and α-linolenic acids (18:3) are the main polyunsaturated fatty acids in plant lipids and play key roles in plant metabolism, development and the stress response [4]. In higher plants, fatty acid biosynthesis starts in the plastids. The stearoyl acyl carrier protein (ACP) is desaturated by Δ9 stearoyl-ACP desaturase to produce oleoyl-ACP, the main product of plastidial fatty acid biosynthesis [5]. Oleic acid (18:1) is then converted into glycerolipids inside or outside of the plastids and can be further desaturated to 18:2 by the action of Δ^12^ fatty acid desaturases (FADs). The Δ^12^ FADs include the microsomal oleate desaturases (e.g., FAD2), which are located in the endoplasmic reticulum (ER), and plastidial oleate desaturases (e.g., FAD6), which are located in the chloroplast. Microsomal oleate desaturases use phospholipids as acyl substrates and nicotinamide adenine dinucleotide-hydrogen (NADH), NADH-cytochrome b_5_ reductase and cytochrome b_5_ as their electron donor system, whereas plastidial oleate desaturases primarily use glycolipids as acyl carriers and nicotinamide adenine dinucleotide phosphate-hydrogen NAD(P)H, ferredoxin-NAD(P) reductase and ferredoxin as their electron donor system [6].

Plant FADs are regulated by different environmental stressors. Temperature can regulate FADs in plants and is a major environmental factor. Additionally, the regulation of gene expression appears to vary among different species, tissues and genes. Previous studies have demonstrated that cold conditions can increase the polyunsaturated fatty acid content in plants to maintain the fluidity of biological membranes [7]. In olive fruits [8], avocado fruits and cotton cotyledons [9,10], the expression of *FAD2* genes is upregulated, and the amount of the corresponding desaturase protein changes under cold stress. Posttranscriptional mechanisms also regulate desaturase activity, for example, 18:2 content in Arabidopsis cultures increases significantly under cold stress; however, no such differences in *FAD2* gene expression levels were observed [11].

Salt stress, another environmental stressor, can also induce changes in fatty acid composition, and many FADs participate in this process [12]. In Arabidopsis, FAD2 was verified as an important enzyme for salt tolerance, and the *FAD6* gene is also an important component of the plant response to salt stress [13,14]. Additionally, the heterologous expression of sunflower *FAD2-1* or *FAD2-3* in yeast yielded higher levels of 18:2 and increased yeast cell tolerance to salt [15]. The *FAD2* gene may play a key role in regulating and maintaining the lipid composition of intracellular membranes, biophysical characteristics and proper functioning of membrane-attached proteins under salt stress conditions [12,13,16,17].

Regarding microsomal oleate desaturases genes, safflower contains at least eleven *FAD2* genes, making it the species with the most *FAD2* gene copy numbers. In yeast, *CtFAD2-1* (KC257447.1), *CtFAD2-2* (KC257448.1), *CtFAD2-10* (KC257447.1) and *CtFAD2-11* (KC257456.1) have been previously verified to exert oleic acid desaturase activity [18]; therefore, in this study, these four *FAD2* genes were used for the expression analysis in further experiments. In terms of plastidial oleate desaturases genes, only a partial sequence of *FAD6* has been cloned [19]_,_ and the characteristics and function of the *FAD6* gene have not yet been studied. In the current study, the plastidial oleate desaturase gene *FAD6* was cloned, its sequence was analyzed and its function was confirmed in cyanobacterium.

Plant development and yield can easily be affected by obvious environmental stressors, namely extreme temperature stress and salt stress. Safflower is a species with strong stress resistance, and previous research revealed that an unsaturated fatty acid may participate in its stress resistance [20]. In this study, the effects of temperature and salt stress on the expression of Δ^12^ fatty acid desaturase genes and fatty acid compositions were examined to determine the factors that regulate 18:2 biosynthesis in safflower. This study systematically explores the responses of Δ^12^ fatty acid desaturase genes to cold, heat and salt at the transcriptional level and provides a theoretical basis for resistant breeding based on fatty acid regulation at the molecular level.

## 2. Results

### 2.1. Cloning and Functional Analysis of the Safflower FAD6 Gene in Cyanobacterium

Before beginning studies investigating the regulation of safflower *FAD2* and *FAD6* gene expression, the *FAD6* gene sequence was cloned and analyzed, and its functional identity was confirmed in cyanobacterium. First, a full-length cDNA clone corresponding to the safflower plastidial oleate desaturase was isolated and designated as *CtFAD6*. A full-length cDNA of 1329 bp that contained an open reading frame (ORF) encoding a predicted protein of 442 amino acid residues was obtained. The sequence analysis revealed a calculated molecular mass of 51.064 kDa and a pI of 9.18. The CtFAD6 amino acid sequence displayed significant similarity to other known plant *FAD6* sequences, exhibiting the highest identity (92%) to the *Cynara cardunculus* var. *scolymus FAD6* gene and an average similarity (75%) to all the analyzed plants. These results suggest that *CtFAD6* encodes a plastidial ω-6 fatty acid desaturase.

Alignment of the FAD6 amino acid sequences of ten different plants revealed three histidine-conserved residue boxes (HDCAH, HDRHH and HIPHH) (Appendix A). These three histidine boxes are characteristic of all membrane-bound FADs and are believed to combine with Fe^2+^ and comprise the catalytic center of the enzyme, since they form ligands to a diiron cluster at the catalytic site [21]. An analysis of the hydrophilicity/hydrophobicity of the CtFAD6 protein with the online tool ProtScale revealed that arginine and isoleucine had the strongest hydrophilicity and hydrophobicity, respectively. In addition, six different hydrophobic regions were found. The second to fifth hydrophobic regions spanned the membrane. An analysis of the CtFAD6 amino acid sequence with targeting prediction tools revealed a 67 bp transit peptide in the N-terminus; its subcellular localization was predicted in the mitochondria, and it belonged to the plastidial FADs (Appendix A).

Finally, a functional analysis of the safflower *FAD6* gene was performed. Since cyanobacteria and higher-plant plastids contain similar lipid and 18:1 fatty acid species and since their desaturases both utilize Fe^2+^ as the immediate electron donor [22], it was reasonable to test the biological function of the cDNA insert in the psyn_6 vector in a cyanobacterium. *Synechococcus* sp. strain PCC7942, which produces saturated and monounsaturated fatty acids but lacks all polyunsaturated fatty acids, was chosen. A chimeric gene was constructed comprising a bacterial psbA promoter fused to the *CtFAD6* coding sequence and cloned adjacent to a spectinomycin selectable marker with the *Synechococcus* sp. strain PCC7942 genomic fragment of a cyanobacterial recombinational transformation vector.

The fatty acid analysis of psyn_6: *CtFAD6*-transformed cyanobacteria cells cultured at a standard growth temperature (34 °C) under continuous illumination showed the presence of 18:2, which was not present in cells transformed with the empty vector (Figure 1). This result indicates that expression of the *CtFAD6* gene was functional: endogenous oleic acid (18:1^Δ9^) was partially transformed into linoleic acid (18:2^Δ9, 12^), endogenous palmitoleic acid (16:1^Δ9^) was partially transformed into a specific fatty acid (16:2^Δ9, 12^) and 16:1 was predominately utilized as a substrate for the FAD6 enzyme. In total, both palmitoleic acid (16:1) and oleic acid (18:1) can be catalyzed to 16:2 and 18:2, respectively, by the CtFAD6 enzyme. *FAD6* homologous genes have been reported to convert 18:1 into 18:2 and also convert 16:1 into 16:2 fatty acids [8]. Our results were consistent with the abovementioned research.

### 2.2. Safflower Oleate Desaturase Genes Are Transcriptionally Regulated by Temperature

To study the effects of temperature on the expression levels of Δ^12^ fatty acid desaturase genes and 18:2 content in safflower tissues, RT-qPCR was performed. The normal growth temperature (16 °C) for safflower seedlings was used as the control temperature. As shown in Figure 2, in roots (Figure 2A) when safflower seedlings were incubated at 4 °C for 12 h, the expression levels of *CtFAD2-2* and *CtFAD2-11* decreased to approximately 0.6-fold, while, at 24 °C for 12 h, the expression level of *CtFAD2-10* increased to 1.35-fold, and at 10 °C, the *CtFAD2* expression level did not increase significantly compared with the expression at 16 °C in roots. In stems (Figure 2B), when safflower seedlings were incubated at 4 °C for 12 h, the *CtFAD2-1* expression level increased to 2.62-fold, and at 24 °C for 12 h, the *CtFAD2-1* expression increased to 3.31-fold, while the *CtFAD2-1* expression was not significantly changed at 10 °C. At low temperatures, specifically 4 °C and 10 °C, the *CtFAD2-10* expression level decreased significantly. In leaves (Figure 2C), expression of the *CtFAD2-1*, *CtFAD2-2*, *CtFAD2-11* and *CtFAD6* genes was obviously changed and was especially intense in the case of *CtFAD6*. At 4°C, *CtFAD2-1*, *CtFAD2-2*, *CtFAD2-11* and *CtFAD6* were significantly induced; the highest expression levels appeared at 4 °C for 1 h (8.39-fold for *CtFAD2-1,* 3.47-fold for *CtFAD2-11*, 8.69-fold for *CtFAD2-2* and 12.15-fold for *CtFAD6*). *CtFAD2-10* was also induced and reached its maximum at 4 °C for 6 h (3.24-fold). At 24 °C for 1 h, the *CtFAD2-1*, *CtFAD2-2*, *CtFAD2-11* and *CtFAD6* genes were significantly induced, while at 10 °C, only *CtFAD2-1* and *CtFAD2-2* slightly increased compared with the expression at 16 °C in leaves.

Overall, the temperature response patterns of Δ^12^ fatty acid desaturase genes showed significant divergences in different tissues. The oleate desaturase genes were significantly induced in safflower leaves, slightly induced in stems and showed extremely weak responses in roots. *CtFAD2-1*, *CtFAD2-2*, *CtFAD2-11* and *CtFAD6* were intensely induced by cold stress and were affected by heat stress.

The fatty acid composition was measured by GC-MS in the different tissues under temperature stress. The results are shown in Table 1. Four fatty acids (stearic acid (SA; 18:0), palmitic acid (PA; 16:0), α-linolenic acid (LnA; 18:3) and linoleic acid (LA; 18:2)) were profiled using GC. No obvious change in the 18:2 content was observed in roots at different temperatures, and the 18:3 content significantly increased at 4 °C for 12 h. The content of 16:0 fatty acid increased significantly, and the 18:0 content decreased significantly under cold stress. In stems, the 18:2 content slightly increased at 4 °C and 24 °C, and at the same temperatures, the 18:3 content significantly increased, the 16:0 content decreased and the 18:0 content did not change significantly. In leaves, at 4 °C and 24 °C, the 18:2, 18:3 and 16:0 contents significantly increased compared with the contents at 16 °C, but the 18:0 content significantly decreased. The above results are in accordance with the expression patterns of Δ^12^ fatty acid desaturase genes at certain temperatures in different tissues, suggesting that the temperature, especially low temperatures, regulates Δ^12^ fatty acid desaturase genes at the transcriptional level and that photosynthetic tissue tends to be more sensitive to temperature than other tissues.

### 2.3. Safflower Oleate Desaturase Genes Are Regulated by Salt

To study the effect of salt on the expression levels of Δ^12^ fatty acid desaturase genes and fatty acid contents in safflower tissues, including roots, stems and leaves, RT-qPCR was performed. The results are shown in Figure 3. The expression of the Δ^12^ fatty acid desaturase genes was barely induced at 0–24 h after treatment with 100 mmol/L salt, while the expressions were significantly induced at 48–72 h after salt treatment. In roots (Figure 3A), *CtFAD2-2*, *CtFAD2-10*, *CtFAD2-11* and *CtFAD6* were significantly induced by salt and reached the maximum expression at 48 h after salt stress (1.88-fold for *CtFAD2-2*, 4.22-fold for *CtFAD2-10*, 4.89-fold for *CtFAD2-11* and 1.85-fold for *CtFAD6*). Additionally, *CtFAD2-1* was highly expressed at 72 h after salt stress (3.34-fold) (Figure 3A). In stems (Figure 3B), *CtFAD2-1*, *CtFAD2-10* and *CtFAD2-11* were induced at 48 and 72 h, and the *CtFAD2-1* expression level reached its maximum at 72 h after stress (7.06-fold), while the *CtFAD2-10* and *CtFAD2-11* expression levels reached their maximum at 48 h after stress (5.71-fold for *CtFAD2-10* and 4.75-fold for *CtFAD2-10*). No obvious changes were found for *CtFAD2-2* and *CtFAD6* after salt stress. In leaves (Figure 3C), *CtFAD2-1*, *CtFAD2-10* and *CtFAD2-11* were significantly induced, and the highest expression levels appeared at 48 h after stress (11.15-fold for *CtFAD2-1*, 9.71-fold for *CtFAD2-10* and 8.20-fold for *CtFAD2-11*). No obvious changes were found for *CtFAD2-2*, and a slight increase was detected for *CtFAD6* at 6 h under salt stress.

To determine whether the fatty acid content was changed by salt stress, four kinds of fatty acids were measured by GC-MS in different tissues at 6, 12, 24, 48 and 72 h after salt stress. As shown in Table 2, in roots, the 18:2 content slightly increased at 24 h, significantly increased at 48 h and reached its maximum at 72 h after salt stress. The content of 18:3 increased and reached its maximum at 48 h after salt stress. No significant change was found for 16:0, and the 18:0 content significantly decreased and reached its minimum at 48 h. In stems, the 18:2 content significantly increased at 48 h and reached its maximum at 72 h after salt stress. The content of 18:3 increased and reached its maximum at 72 h after salt stress. No significant change was found for 16:0, and the 18:0 content significantly decreased and reached its minimum at 72 h. In leaves, the 18:2 content was not significantly different from that of the control, while the salt treatment significantly increased the relative percentage of 18:3 at 24 h after salt stress (Table 2). Salt stress significantly increased the expression of some Δ^12^ fatty acid desaturase genes, while the 18:2 content was not induced dramatically by salt. However, the downstream product, 18:3, was increased significantly, and the 18:0 content decreased significantly. Thus, more 18:2 may be consumed as a substrate for 18:3 biosynthesis.

## 3. Discussion

Some Δ^12^ fatty acid desaturase genes were affected by both cold and heat stress, and the corresponding 18:2 content was also changed. Plants can usually increase the content of unsaturated fatty acids in membrane lipids to enhance their cold resistance, although for different plants, the effects of different temperatures on the expression of Δ^12^ fatty acid desaturase genes are divergent. For instance, *PfFAD2a/b* transcription is differentially induced by cold (4 °C) and repressed by heat (42 °C) in chia and perilla [23,24]. A similar phenomenon was observed in ginkgo, in which *GbSAD* and *GbFAD2* transcripts in leaves were induced by cold stress (4 °C, 15 °C) but inhibited by high temperatures (35 °C, 45 °C) [25]. However, unlike perilla, chia and ginkgo, *PlFAD2* was upregulated by high temperature in cold-sensitive materials in lima bean leaves [26]. The unsaturated fatty acid content in safflower tissues increased significantly at 5 °C, and the corresponding saturated fatty acid contents decreased [19]. In our study, the safflower *FAD2* gene, which systematically responds to different temperatures, was first explored. In general, 4 °C significantly induced the expression of Δ^12^ fatty acid desaturase genes (*FAD2* and *FAD6*), while the Δ^12^ fatty acid desaturase genes were slightly induced by 10 °C and partially induced by 24 °C. However, this temperature response pattern of the Δ^12^ fatty acid desaturase genes showed obvious divergences among different tissues and was primarily induced in leaves. This phenomenon was also observed in the research by Guan LL [19]. Generally, Δ^12^ fatty acid desaturase genes in most plants, including safflower, chia, perilla and gingko, are induced by the temperature, mainly in leaves. One reason may be that seedling leaves are more sensitive to temperature changes, while the root temperature is relatively stable under stress conditions. Additionally, the different fatty acid compositions in various plant tissues may cause divergent response patterns to the temperature. Furthermore, some researchers have found that the *FAD2* gene shows obvious light dependence on the regulatory response to cold stress in cotton [10]; this finding also explains why the *CtFAD2* gene is unregulated primarily and preponderantly in photosynthetic tissues. The light-dependent response could be due to an indirect hormonal effect or to the direct effect of light-regulatory elements on the *CtFAD2-1* and *CtFAD2-2* promoters.

Salt stress is also a key factor that affects the expression of fatty acid desaturase genes and unsaturated fatty acid composition. Previous research revealed that *PlFAD2* was upregulated by salt stress in lima beans [26]. In this study, in contrast to the temperatures at which genes were rapidly induced, the Δ^12^ fatty acid desaturase genes were not obviously induced by salt at the early stage, and *CtFAD2-1*, *CtFAD2-10* and *CtFAD2-11* were upregulated after 2 d after salt stress. In addition, unlike temperature stress, the expression of the Δ^12^ fatty acid desaturase genes was induced by salt in all detected safflower tissues (roots, stems and leaves). We speculate that long-term exposure to a highly osmotic environment caused cellular dehydration, and some plant hormones may be induced by this dehydration phenomenon [12]. However, it will be necessary to explore and verify these results in subsequent studies. Although the 18:2 content in the leaves did not significantly change after the salt treatment, the 18:3 content increased significantly, consistent with the increased expression levels of some Δ^12^ fatty acid desaturase genes. These results indicate that the regulation of temperature and salt stress to Δ^12^ fatty acid desaturase genes partially at the transcriptional level. However, which trans factors are involved in regulation under these stressors and their interactions and connections should be further explored in a future study.

## 4. Conclusions

In the present work, by performing functional expression in cyanobacteria, we demonstrated that the safflower *FAD6* gene encodes a plastidial oleate desaturase. Furthermore, transcriptional regulation of the oleate desaturase genes in safflower under temperature and salt stress revealed that temperature, especially low temperatures, can regulate the expression of Δ^12^ fatty acid desaturase genes, and it seems as though the photosynthetic tissue tends to be more sensitive to temperature than other tissues; salt stress can induce the upregulated expression of Δ^12^ fatty acid desaturases genes, except *CtFAD2-2* and *CtFAD6* in different tissues. This study constitutes a significant step towards an understanding of the factors involved in the regulation of 18:2 biosynthesis in safflower and its adaptive mechanism under stress.

## 5. Materials and Methods

### 5.1. Plant Material and Stress Treatments

A *Carthamus tinctorius* L. (Asteraceae) cultivar (Chuanhong No. 1) was selected for this experiment. Safflower seedlings were incubated as a control group in an illumination incubator at 16 °C (the general growth temperature of safflower seedlings in a field in Sichuan) under a photoperiod of 16:8 h (light:dark) and a light intensity of 45 μmol m^−2^ s^−1^ diffuse light. After three leaf stages, the partial seedlings were transferred to 4 °C, 10 °C and 24 °C, and the illumination conditions were the same as those for the control group. The partial safflower seedlings were then transferred into a matrix containing vermiculite and quartz sand (3:1) and watered by Hoagland’s nutrient solution. A progressive increase in the salt concentration was applied using a solution of 100 mmol/L NaCl to avoid the effects of salt shock.

For temperature stress, safflower tissues (roots, stems and leaves) were obtained 0, 1, 6 and 12 h after stress. For salt stress, the same tissues were obtained 0, 6, 12, 24, 48 and 72 h after the end of the progressive increase in salt concentration. The consistent plants were selected and immediately frozen in liquid nitrogen after harvesting and stored at −80 °C until total RNA was isolated. All experiments were performed with three biological replicates.

### 5.2. Total RNA Extraction and cDNA Synthesis

Total RNA was isolated from *Carthamus tinctorius* L. tissues (roots, stems and leaves) using a Plant RNA Extraction Kit (Tiangen, Beijing, China). First-stranded cDNA was synthesized from 1 μg of DNase-treated RNA using TaKaRa reverse transcription reagents following the manufacturer’s instructions (TaKaRa Bio, Dalian, China). cDNA was stored at −20 °C.

### 5.3. Isolation of a Full-Length Plastidial Oleate Desaturase cDNA Clone from Safflower

Primers (PF1 and PR1) were designed according to the transcription data (Appendix A). PCR amplification was then performed. A full-length DNA fragment of the expected size was generated and subcloned into the pclone007-T vector (TSINGKE, Beijing, China), and the plasmids were extracted and then sequenced. The DNA sequence was analyzed with the LASERGENE software package (DNASTAR, Madison, WI, USA), and the multiple sequence alignments of the FAD6 amino acid sequences were calculated using the ClustalX program and displayed with GeneDoc. A phylogenetic tree analysis was performed using the neighbor-joining method implemented in the Phylip package using Kimura’s correction for multiple substitutions and a 1000 bootstrap data set. Subcellular localization was predicted using two different programs: Wolf PSORT (http://wolfpsort.org/, accessed on 19 December 2019) and TargetP (http://www.cbs.dtu.dk/services/TargetP/, accessed on 19 December 2019). Signal peptides were predicted using SignalP 4.0 (http://www.cbs.dtu.dk/services/SignalP/, accessed on 19 December 2019).

### 5.4. Quantitative Real-Time PCR (qRT-PCR)

Gene expression analysis was performed by qRT-PCR as previously described [27]. qRT-PCR was carried out by using the SYBR Green I Master Mix (Tiangen, Beijing) with three replicates. Specific primers (PF2/PR2, PF3/PR3, PF4/PR4, PF5/PR5, PF6/PR6 and PF7/PR7) were designed by Primer 3.0 (Appendix A). The housekeeping genes *EF1* and *UBCE2* were used as endogenous reference genes for normalization [28]. The amplification specificity of all desaturase gene-specific and reference gene-specific primers was confirmed by observing a single dissociation curve for each pair of primers.

Each 10 μL reaction contained 5 μL of SYBR Green Master Mix, 50 ng of cDNA and forward and reverse primers each at 5 pmol. The first denaturation step for 90 s at 94 °C was followed by 35 cycles of 10 s at 94 °C, 5 s at 55 °C and 10 s at 72 °C. The fluorescence was measured three times at the end of the extension step at 72 °C, 81 °C and 84 °C, followed by a final extension step of 5 min at 72 °C. Additionally, it was ensured that the amplification efficiencies of the target and reference genes were similar. In this study, the amplification efficiencies of the reference gene and target genes were all between 90% and 110%, and the standard deviations of the Cq values were all less than 0.2. Finally, the 2^−ΔΔCt^ method was used to express the target gene results relative to the reference gene results [29]. Data are presented as the mean ± standard deviation (SD) of three reactions performed in different 96-well plates, each with three replicates per plate.

### 5.5. Expression and Sequence Analysis of the Safflower CtFAD6 Gene in Cyanobacterium

A primer with a homologous arm was designed based on the pysn_6 vector (PF8/PR8), and the sequence with the homologous arm was cloned. The pysn_6 vector was digested with the HindIII and BamHI restriction enzymes. The *CtFAD6* fragment and digested vector were connected by homologous recombinase (Vazyme, Nanjing, China). The recombinant vector and empty vector were transformed in *Synechococcus* sp. strain PCC7942 using the natural transformation method described by Golden [30]. Transformants were placed onto BG-11 plates containing 10 mg/L spectinomycin and incubated at 34 °C under light. After 1~2 weeks, several resistant transformants appeared as green colonies, and several positive colonies were picked and re-streaked onto 50 mL of fresh medium containing spectinomycin. When the strains were in the log phase, the bacteria were centrifuged and collected for the subsequent analysis.

### 5.6. Fatty Acid Analysis

Cyanobacteria cells were harvested by centrifugation at 5000× *g* for 10 min at 4 °C and washed three times with distilled water. The pellets were then dried. The plant samples (0.3 g) were weighed and used for the subsequent analysis. The cyanobacteria cells and plant samples were transmethylated, extracted and analyzed by a gas chromatography/mass spectrometry (GC/MS) system using an Agilent 5973 MS system coupled with an Agilent 6890 gas chromatograph fitted with an HP-SMS capillary column (30 × 0.25 mm, film thickness 0.25 µm) [31,32]. Helium was used as the carrier gas at a flow rate of 1 mL min^−1^ with an injector temperature of 250 °C, a split ratio of 50:1 and a temperature program of 120 °C to 220 °C at a rate of 4 °C min^−1^.

### 5.7. Statistical Analysis

The experiment was performed with a completely randomized design with three replicates. Analysis of variance was performed to assess the effects of temperature and salt stress on the fatty acid relative content percentages in three safflower tissues. Data were analyzed, and the means were compared by Duncan’s multiple range tests using the SPSS program ver. 20.

## Figures and Tables

**Figure 1 ijms-24-02765-f001:**
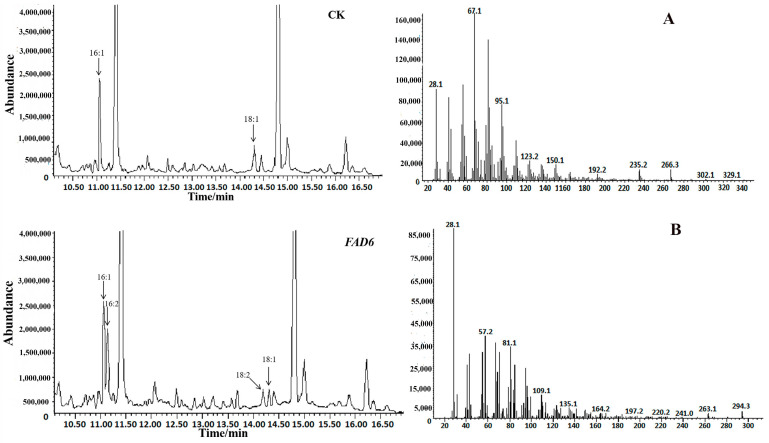
GC analysis of FAMEs from the cyanobacterium cells transformed with the plasmid psyn-*CtFAD6* and empty vector (CK). Note: The (**A**,**B**) are the spectrum of peaks 16:2 and 18:2, respectively.

**Figure 2 ijms-24-02765-f002:**
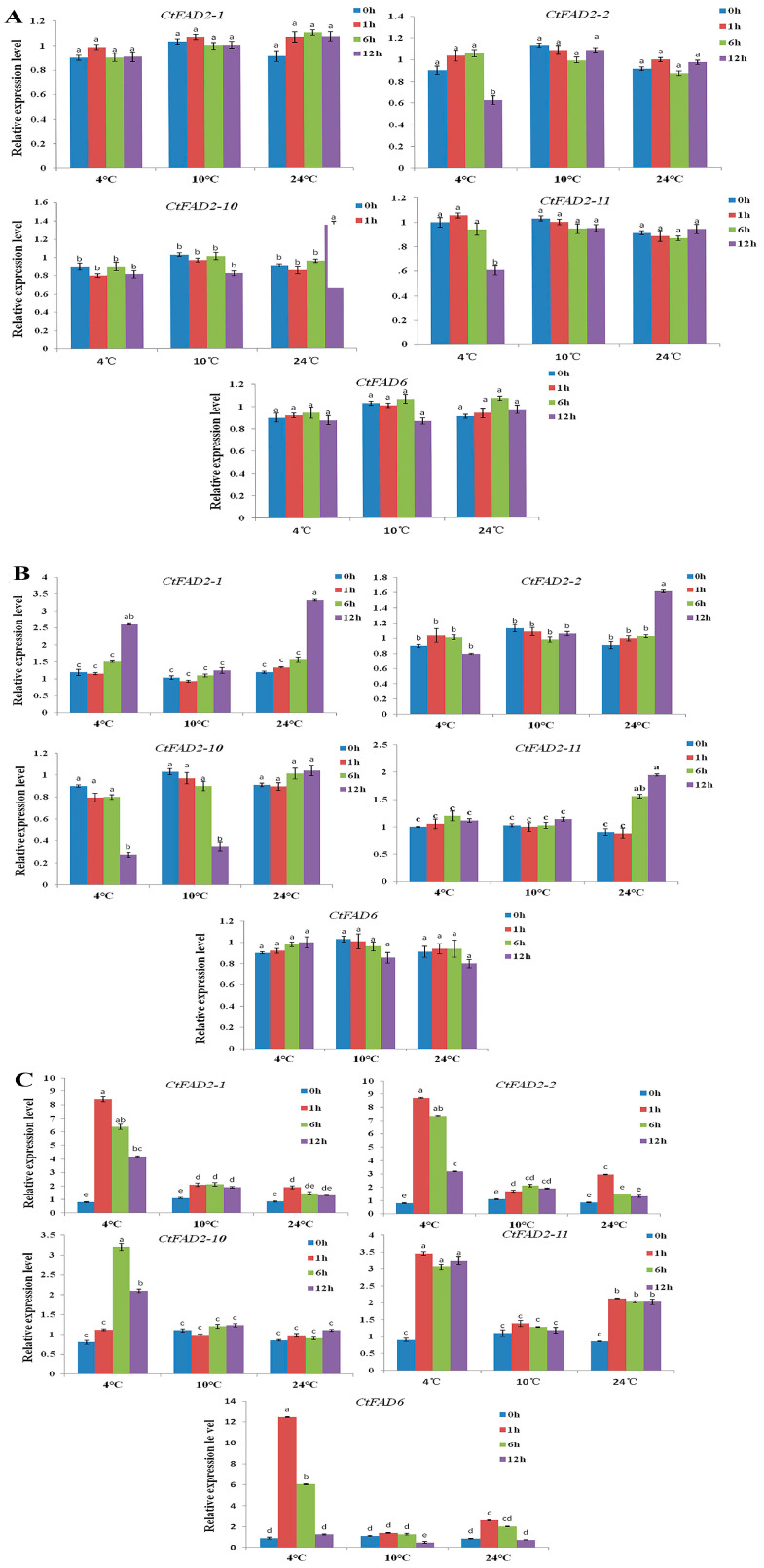
Effect of different temperatures on the relative expression levels of safflower *CtFAD2-1*, *CtFAD2-2*, *CtFAD2-10*, *CtFAD2-11* and *CtFAD6* in roots, stems and leaves, and the expression levels were detected at 0, 1, 6 and 12 h after temperature stress. Note: The letter “(**A**)” represents the roots, the letter “(**B**)” represents the stems and the letter “(**C**)” represents the leaves. The transcription levels for three tissues were normalized against the samples under 16 °C, respectively. Each bar represents the mean. Error bars represent the standard deviation of the mean (*n* = 3). The different alphabets above each bar indicate significant differences (*p* < 0.05).

**Figure 3 ijms-24-02765-f003:**
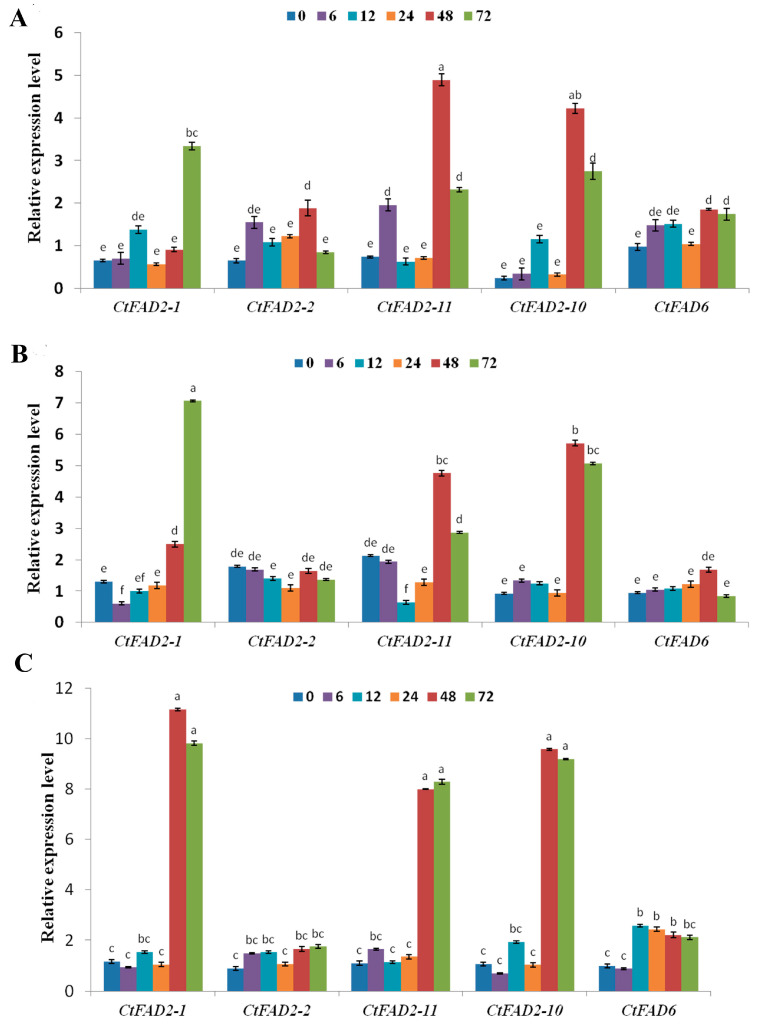
Effect of salt on the relative expression levels of safflower *CtFAD2-1*, *CtFAD2-2*, *CtFAD2-10*, *CtFAD2-11* and *CtFAD6* genes in root, stem and leaf tissues; the expressions were detected at 0, 6, 12, 24, 48 and 72 h after salt stress treatment. Note: The letter “(**A**)” represents the roots, the letter “(**B**)” represents the stems and the letter “(**C**)” represents the leaves. Transcription levels for the three tissues were normalized against the samples treated without salt stress, respectively. Each bar represents the mean. Error bars represent the standard deviation of the mean (*n* = 3). The different alphabets above each bar indicate significant differences (*p* < 0.05).

**Table 1 ijms-24-02765-t001:** Effect of different temperatures on the composition and contents of fatty acids in safflower different tissues.

Roots (%)										
Fatty acids	NT (16 °C)	4 °C—1	4 °C—6	4 °C—12	10 °C—1	10 °C—6	10 °C—12	24 °C—1	24 °C—6	24 °C—12
Linoleic acid (18:2)	4.306 ± 0.144 a	4.510 ± 0.190 a	4.514 ± 0.121 a	4.676 ± 0.184 a	4.353 ± 0.193 a	4.321 ± 0.113 a	4.597 ± 0.177 a	4.467 ± 0.158 a	4.090 ± 0.193 a	4.282 ± 0.875 a
Linolenic acid (18:3)	2.969 ± 0.321 b	3.337 ± 0.154 ab	3.614 ± 0.231 ab	4.368 ± 0.120 a	3.007 ± 0.156 b	2.684 ± 0.032 b	3.162 ± 0.188 b	3.181 ± 0.056 b	2.567 ± 0.012 b	2.951 ± 0.024 b
palmitic acid (16:0)	53.953 ± 0.022 b	61.275 ± 0.026 a	60.066 ± 0.009 a	60.658 ± 0.005 a	61.212 ± 0.165 a	61.746 ± 0.165 a	61.087 ± 0.199 a	61.549 ± 0.078 a	61.258 ± 0.482 a	61.297 ± 0.320 a
stearic acid (18:0)	38.773 ± 0.175 a	30.879 ± 0.018 b	31.807 ± 0.568 b	30.109 ± 0.062 b	30.934 ± 0.224 b	31.588 ± 0.226 b	31.072 ± 0.210 b	30.804 ± 0.261 b	31.086 ± 0.889 b	31.471 ± 0.124 b
Stems (%)										
Fatty acids	NT (16 °C)	4 °C—1	4 °C—6	4 °C—12	10 °C—1	10 °C—6	10 °C—12	24 °C—1	24 °C—6	24 °C—12
Linoleic acid (18:2)	4.284 ± 0.185 ab	4.897 ± 0.097 a	4.690 ± 0.253 a	4.988 ± 0.227 a	3.79 ± 0.281 b	4.229 ± 0.237 ab	4.572 ± 0.193 a	4.433 ± 0.164 a	4.88 ± 0.056 a	4.919 ± 0.187 a
Linolenic acid (18:3)	2.474 ± 0.077 b	3.414 ± 0.077 a	3.258 ± 0.262 a	3.549 ± 0.056 a	2.926 ± 0.281 ab	2.810 ± 0.195 ab	2.707 ± 0.322 ab	2.664 ± 0.617 ab	4.038 ± 0.356 a	3.984 ± 0.170 a
palmitic acid (16:0)	61.869 ± 0.056 a	60.805 ± 0.232 b	60.997 ± 0.256 b	60.809 ± 0.003 b	62.237 ± 0.233 a	62.286 ± 0.002 a	61.961 ± 0.310 a	62.585 ± 0.352 a	60.777 ± 0.338 b	60.870 ± 0.586 b
stearic acid (18:0)	30.727 ± 0.354 a	30.885 ± 0.321 a	31.055 ± 0.322 a	30.655 ± 0.008 a	31.097 ± 0.552 a	30.576 ± 0.114 a	30.761 ± 0.284 a	30.619 ± 0.214 a	30.443 ± 0.258 a	31.030 ± 0.362 a
Leaves (%)										
Fatty acids	NT (16 °C)	4 °C—1	4 °C—6	4 °C—12	10 °C—1	10 °C—6	10 °C—12	24 °C—1	24 °C—6	24 °C—12
Linoleic acid (18:2)	5.009 ± 0.079 c	7.602 ± 0.058 a	5.980 ± 0.237 b	7.363 ± 0.346 a	4.678 ± 0.232 c	5.558 ± 0.213 bc	5.597 ± 0.177 bc	6.011 ± 0.071 b	5.836 ± 0.075 b	6.862 ± 0.224 ab
Linolenic acid (18:3)	20.575 ± 0.321 c	23.044 ± 0.612 a	23.434 ± 0.234 a	22.217 ± 0.131 ab	17.669 ± 0.543 d	20.859 ± 0.320 c	17.618 ± 0.156 d	22.961 ± 0.231 a	23.009 ± 0.304 a	23.558 ± 0.125 a
palmitic acid (16:0)	45.722 ± 0.889 d	45.856 ± 0.907 d	47.649 ± 0.256 c	47.215 ± 0.078 c	52.384 ± 0.127 a	48.996 ± 0.482 c	50.716 ± 1.26 b	48.161 ± 0.226 c	48.106 ± 0.178 c	47.739 ± 0.056 c
stearic acid (18:0)	26.514 ± 0.299 a	23.500 ± 0.470 c	22.938 ± 0.455 c	23.206 ± 0.019 c	25.570 ± 0.119 ab	26.388 ± 0.889 ab	26.071 ± 0.262 a	22.869 ± 0.084 c	22.864 ± 0.300 c	22.842 ± 0.124 c

Note: The units were the relative percentage content (%) of fatty acids. 4 °C—1, 4 °C—6, 4 °C—12, 10 °C—1, 10 °C—6, 10 °C—12, 24 °C—1, 24 °C—6 and 24 °C—12 refer to 4 °C for 1 h, 4 °C for 6 h, 4 °C for 12 h, 10 °C for 1 h, 10 °C for 6 h, 10 °C for 12 h, 24 °C for 1 h, 24 °C for 6 h and 24 °C for 12h, respectively. NT (16 °C) refers to the samples cultivated under normal temperature (16 °C) as the control, and the different letters indicate significant differences at the level 0.01 (*p* ≤ 0.01).

**Table 2 ijms-24-02765-t002:** Effect of salt on the compositions and contents of fatty acids in safflower different tissues.

Roots (%)						
Fatty acids	CK	Salt—6 h	Salt—12 h	Salt—24 h	Salt—48 h	Salt—72 h
Linoleic acid (18:2)	2.259 ± 0.088 b	2.250 ± 0.025 b	2.212 ± 0.002 b	2.738 ± 0.047 ab	3.573 ± 0.332 a	3.602 ± 0.049 a
Linolenic acid (18:3)	1.038 ± 0.024b	1.012 ± 0.009 b	1.038 ± 0.007 b	1.445 ± 0.008 ab	1.937 ± 0.019 a	1.241 ± 0.002 b
palmitic acid (16:0)	58.407 ± 0.013 a	58.056 ± 0.012 a	58.010 ± 0.137 a	57.972 ± 0.010 a	58.942 ± 0.007 a	58.515 ± 0.029 a
stearic acid (18:0)	38.296 ± 0.006 a	38.682 ± 0.177 a	38.74 ± 0.021 a	37.415 ± 0.002 ab	37.078 ± 0.006 b	37.642 ± 0.004 b
Stems (%)						
Fatty acids	CK	Salt—6 h	Salt—12 h	Salt—24 h	Salt—48 h	Salt—72 h
Linoleic acid (18:2)	3.056 ± 0.200 b	3.122 ± 0.087 b	3.800 ± 0.061 ab	3.707 ± 0.185 ab	4.033 ± 0.065 a	4.205 ± 0.046 a
Linolenic acid (18:3)	1.297 ± 0.332 b	1.255 ± 0.133 b	1.219 ± 0.081 b	2.254 ± 0.066 a	2.450 ± 0.016 a	2.576 ± 0.018 a
palmitic acid (16:0)	57.529 ± 0.000 a	57. 004 ± 0.009 a	57.027 ± 0.008 a	57.890 ± 0.001 a	57.370 ± 0.013 a	56.776 ± 0.003 ab
stearic acid (18:0)	38.118 ± 0.008 a	38.619 ± 0.007 a	37.954 ± 0.173 a	36.149 ± 0.058 b	36.147 ± 0.010 b	36.443 ± 0.002 b
Leaves (%)						
Fatty acids	CK	Salt—6 h	Salt—12 h	Salt—24 h	Salt—48 h	Salt—72 h
Linoleic acid (18:2)	5.263 ± 0.173 a	5.168 ± 0.055 a	5.566 ± 0.250 a	5.325 ± 0.606 a	5.503 ± 0.183 a	5.038 ± 0.166 a
Linolenic acid (18:3)	16.262 ± 0.107 c	16.986 ± 0.157 c	16.563 ± 0.065 c	22.794 ± 0.359 a	20.703 ± 0.369 b	20.750 ± 0.262 b
palmitic acid (16:0)	50.259 ± 0.008 a	47.663 ± 0.004 ab	49.556 ± 0.002 a	43.544 ± 0. 008 c	49.336 ± 0.025 a	50.716 ± 0.009 a
stearic acid (18:0)	29.063 ± 0.012 ab	30.183 ± 0.002 a	28.315 ± 0. 008 b	28.337 ± 0.001 b	22.458 ± 0.024 c	23.496 ± 0.889 c

Note: The units were the relative percentage content (%) of fatty acids. Salt—6 h, Salt—12 h, Salt—24 h, Salt—48 h and Salt—72 h refer to the samples were collected at 6, 12, 24, 48 and 72 h after salt stress, respectively. CK refers to the samples cultivated under normal condition as the control, and the different letters indicate significant differences at the level 0.01 (*p* ≤ 0.01).

## Data Availability

Not applicable.

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
