# Peer review of "Effects of Temperature and Salt Stress on the Expression of delta-12 Fatty Acid Desaturase Genes and Fatty Acid Compositions in Safflower"

_ijms, 2023, doi:10.3390/ijms24032765_

Round 1

Reviewer 1 Report

The manuscript by Li et al., describes that two abiotic stresses (salt and temp) can significantly but differentially regulate the desaturase gene expression and fatty acid compositions in the tissues of safflower. QPCR was used to study gene expressions and GCMS was used to study the fatty acid composition. I think it would have been better if they had gone with a transcriptomic approach rather than qPCR (with just a limited number of genes). Having said that the current methodology used in the manuscript is sufficient for the message detailed in the manuscript. I have minor comments below.

  Line 10: Since you never mentioned second in the abstract remove, “first”.

Line 41: Whereas plastidial oleate desaturases primarily use glycolipids as acyl carriers and nicotinamide adenine ……

Line 70: have been previously verified to exert

Line 78: temperature, drought, and salt

Line 98: spacing

Fig. 1 and Fig. 2. .3 are blurred. Please change the images.

Line 118: remove both

Table 2 should move before the discussion section

Line 47: Additionally, the different fatty acid compositions in various plant tissues may cause divergent response patterns to temperature.

Line 64 upregulated after 2 d of salt stress.

Line 101: Are all samples of stem, root, and leaves collected from the same plant or different plants?

What primers were used for C-DNA synthesis?

Reviewer 2 Report

In the present manuscript the authors try to drwa conclusions between the expression of selected fatty acid desaturase (FAD) genes and the fatty acid composition of safflower plants. In the first part of the study they clone a previously described putative plastidial FAD6 homolog. By expressing it in cyanobacteria its enzymatic activity could be shown. This part of the manuscript is quite straight although it was not clear how the sequence was deduced or on which basis the primers were designed. Additionally, it would have been far more convincing to make a CtFAD6-GFP fusion and to express it in tobacco to show plastidial localization instead of in silico prediction.

In the second, larger, part of the study the authors performed quantitative RT-PCR studies and measured relative contents of different fatty acids after different temperature treatments and salt stress. Although the presented data appear sound the conclusions from the correlation between the data sets cannot be drawn in the way as the authors did. Especially, in leave samples some of the analyzed FADs are induced by up to 5-10 fold. But these differences are by far not reflected in fatty acid measurements, where the greatest differences between different treatments in particular tissues were around 50%. A major weakness of the presented approach is that a subset of FAD genes were analyzed and correlated with fatty acid composition, not its absolute contents! Undoubted, there are some significant changes coinciding in both data sets, but significance and correlation are not the same thing. Additionally, instead of lots of graphs and tables, which I found almost impossible to follow, some correlation analyses would have been useful. In this way, at least some of the drawn conclusions could have been supported. But in the current form the manuscript is too descriptive to be published and the conclusions cannot be drawn in the presented way.

Minor points of criticism:

- Shift Fig. 1 to Supplement
- Caption of Fig. 1 is missing acronyms of species and accessions of sequences
- Fig. 3 is not readable, all writings too small

Round 2

Reviewer 2 Report

Now it´s OK, thanks.